# Recent Advancements and Biotechnological Implications of Carotenoid Metabolism of *Brassica*

**DOI:** 10.3390/plants12051117

**Published:** 2023-03-02

**Authors:** Lichun Shi, Lin Chang, Yangjun Yu, Deshuang Zhang, Xiuyun Zhao, Weihong Wang, Peirong Li, Xiaoyun Xin, Fenglan Zhang, Shuancang Yu, Tongbing Su, Yang Dong, Fumei Shi

**Affiliations:** 1School of Life Sciences, Liaocheng University, Liaocheng 252059, China; 2Beijing Vegetable Research Center (BVRC), Beijing Academy of Agriculture and Forestry Science (BAAFS), Beijing 100097, China; 3National Engineering Research Center for Vegetables, Beijing 100097, China; 4Key Laboratory of Biology and Genetic Improvement of Horticultural Crops (North China), Ministry of Agriculture, Beijing 100097, China; 5Beijing Key Laboratory of Vegetable Germplasm Improvement, Beijing 100097, China; 6Marine Science Research Institute of Shandong Province, Qingdao 266104, China; 7State Key Laboratory of Systematic and Evolutionary Botany, Institute of Botany, The Chinese Academy of Sciences, Beijing 100093, China; 8China National Botanical Garden, Beijing 100093, China

**Keywords:** *Brassica*, carotenoid, QTL, genetics, biotechnological implications

## Abstract

Carotenoids were synthesized in the plant cells involved in photosynthesis and photo-protection. In humans, carotenoids are essential as dietary antioxidants and vitamin A precursors. *Brassica* crops are the major sources of nutritionally important dietary carotenoids. Recent studies have unraveled the major genetic components in the carotenoid metabolic pathway in *Brassica*, including the identification of key factors that directly participate or regulate carotenoid biosynthesis. However, recent genetic advances and the complexity of the mechanism and regulation of *Brassica* carotenoid accumulation have not been reviewed. Herein, we reviewed the recent progress regarding *Brassica* carotenoids from the perspective of forward genetics, discussed biotechnological implications and provided new perspectives on how to transfer the knowledge of carotenoid research in *Brassica* to the crop breeding process.

## 1. Introduction

Carotenoids are red, orange, and yellow pigments that are widely distributed in nature, mainly comprising C40 isoprenes. Studies have shown that lutein and β-carotene are the main components of carotenoids, which also include zeaxanthin, cryptoxanthin, astaxanthin, and lycopene [1]. Carotenoids have important functions in plant photosynthesis and are auxiliary pigments in light absorption. Carotenoid molecules contain multiple conjugated double bonds that can absorb energy and protect plants from reactive oxygen species [2]. Carotenoids endow fruit and flowers with bright colors [3], enabling them to attract the attention of pollinating insects and animals. In addition to their roles in plants, carotenoids are vital for the health and nutrition of humans. People take carotenoids to supplement vitamin A, which is used to treat night blindness [4]. Lutein, as a main component of carotenoids, can protect human vision to a certain extent, prevent vision deterioration, and prevent cataracts, and other eye diseases [5]. Astaxanthin also helps the body fight inflammation and boosts immunity [6]. There have been significant efforts to increase the carotenoid content in agricultural crops to improve their nutritional value and health benefits. Manipulating carotenoid metabolism using biotechnology and genetic engineering has been successfully implemented in many crops, and golden rice is the most relevant example of improving β-carotene in food [7].

The metabolism of carotenoids in plants has been the subject of extensive research because of the significance of carotenoids to both plants and humans. The genus *Brassica* includes many vegetable crops, such as *B. rapa* (AA genome), *B. nigra* (BB genome), *B. oleracea* (CC genome), *B. juncea* (AABB genome), *B. napus* (AACC genome), and *B. carinata* (BBCC genome) [8]. These species contain diverse carotenoid metabolites and high levels of nutritionally significant components. Genus and species, as well as genotype and agricultural conditions, affect the carotenoid composition and content of these organisms. The present review focuses on current situation and recent advances in carotenoid metabolism of *Brassica* crops from the perspective of forward genetics. Moreover, we discuss the evolution of carotenoid composition and certain key carotenoid metabolism genes during the formation of the *Brassica* crops. Lastly, we review the agricultural application and the prospects of carotenoid metabolism and its genetic manipulation in *Brassica*.

## 2. Overview of the Metabolic Pathway of Carotenoids

### 2.1. Carotenoid Biosynthesis

Carotenoid biosynthesis pathway components in higher plants have been gradually clarified through biochemical analysis, classical genetics, and molecular genetics. In plants, the methylerythritol phosphate (MEP) pathway is the major carotenoid production route (Figure 1). It can produce the isopentenyl diphosphate (IPP) and its allyl isomer dimethylallyl diphosphate (DMAPP) [9], which are used as substrates to synthesize the C20 geranylgeranyl diphosphate (GGPP) [10]. GGPP is the most direct precursor of plant carotenoids and participates in the synthesis of the earliest carotenoid in plants [11]. The first compound in the carotenoid biosynthesis pathway, 15-cis-phytoene, is synthesized by the condensation of two GGPP molecules, catalyzed by phytoene synthase (PSY) [12]. Phytoene desaturase (PDS) and ζ-carotene desaturase (ZDS) catalyze a four-step dehydrogenation reaction that produces 9,15,9′-cis-ζ-carotene, which is then isomerized into yellow 9,9′-cis-ζ-carotene by ζ-isomerase (ZISO) [13]. Subsequently, carotenoid isomerase (CRTISO) converts yellow prolycopene into red all-trans lycopene [14]. If lycopene β-cyclase (LCYB) acts on both ends of the lycopene molecule, β-carotene is formed. If one of the ends of the lycopene molecule is affected by lycopene ε-cyclase (LCYE), δ-carotene is formed [15]. Then, δ-carotene can be catalyzed by LYCB into α-carotene. Cytochrome P450 carotene β-hydroxylase (CYP97A) and cytochrome P450 carotene ε-hydroxylase (CYP97C) then catalyze α-carotene to create lutein [16]. In addition, β-carotene hydroxylase (BCH) catalyzes β-carotene to produce β-cryptoxanthin, which is converted into zeaxanthin. Zeaxanthin can be further converted to antherxanthin and then to violaxanthin, both of which are catalyzed by zeaxanthin epoxidase (ZEP) [17]. Violaxanthin can also be reversed to form zeaxanthin under the catalysis of violaxanthin de-epoxidase (VDE) [18], via a process termed the xanthophyll cycle [19], which protects plants from light damage. Neoxanthin is then produced from violaxanthin under the catalysis of neoxanthin synthase (NXS) [20].

### 2.2. Degradation of Carotenoids

Carotenoid cleavage dioxygenases (CCDs) lyse carotenoids to form products that provide leaves, flowers, and fruit with specific colors and flavors, and produces abscisic acid (ABA) and other plant hormones. A group of enzymes known as carotenoid cleavage oxygenases (CCOs) catalyze the particular enzymatic oxidative degradation of carotenoids. The varieties of apocarotenoid breakdown products are determined by the precise bonds at which CCOs cleave the carotenoids, with substrate specificity. The 9-cis-epoxycarotenoid dioxygenases (NCEDs) and carotenoid cleavage dioxygenases make up the plant CCO family (which are CCDs) [21]. CCDs are divided into a number of subfamilies, including CCD1, CCD2, CCD4, CCD7, and CCD8. Among them, CCD1 is mainly involved in the formation of plant aromatic substances, and can split β-carotene into β-ionone, an important aroma component [22]. CCD4 is mainly involved in the formation of pigment substances in flowers and fruit. The enzymes encoded by CCD7 and CCD8 are located in plastids and are mainly involved in the synthesis of strigolactone. NECDs are rate-limiting enzymes controlling the production of ABA from carotenoids, which mainly comprises cleaving violaxanthin or neoxanthin, thereby forming precursors of ABA [23]. In addition to the precise cleavage mediated by CCDs, nonspecific enzymes (lipoxygenases and peroxidases) and photochemical oxidation contribute to carotenoid degradation. Carotenoid nonspecific oxidation produces unspecific apocarotenoid factors via random cleavage.

Genes identified by forward genetic analysis which are involved in carotenoid metabolism in *Brassica* were labelled with grey boxes. Enzymes in the carotenoid metabolic pathway were in dark blue. GA3P, Glyceraldehyde 3-phosphate; IPP, isopentenyl diphosphate; DMAPP, dimethylallyl diphosphate; GGPP, geranylgeranyl diphosphate; DXS, 1-deoxy- D –xylulose 5-phosphate synthase; DXR, 1-deoxy- D -xylulose 5-phosphate reductoisomerase; GGPPS, GGPP synthase; PSY, phytoene synthase; PDS, phytoene desaturase; Z-ISO, ζ- carotene isomerase; ZDS, ζ-carotene desaturase; CrtISO, carotenoid isomerase; LCYE, lycopene ε-cyclase; LCYB, lycopene β-cyclase; BCH, β-carotene hydroxylase; CYP97A, cytochrome P450 carotene β-hydroxylase; CYP97C, cytochrome P450 carotene ε-hydroxylase; ZEP, zeaxanthin epoxidase; VDE, violaxanthin de-epoxidase; NXS, neoxanthin synthase; CCD, carotenoid cleavage dioxygenase; NCED, 9-cis-epoxycarotenoid dioxygenase; OR, ORANGE protein.

## 3. Genetic Study of Carotenoid Accumulation in *Brassica* Crops

*Brassica* carotenoid metabolism is important for the development of flower color, which is exploited for decorative and landscaping use. *Brassica* flowers are generally yellow, but can be dark yellow, orange, milky white, or white. Here, we used *B. rapa* (Chinese cabbage) as an example. The main carotenoids in the yellow petals of Chinese cabbage are violaxanthin and lutein [24], while there is more lutein and β-carotene in the orange petals. β-carotene and lutein contents are regarded as key elements related to the yellow pigmentation in the leaves of Chinese cabbage [25]. The lutein content in the internal leaves of yellow cultivars is greater than that in the internal leaves of orange cultivars. Similar observations were made for the inner yellow leaves compared with the white internal leaves. Collectively, these observations demonstrated that carotenoid profiles are distinctive in petals and leaves, and the carotenoid contents of Chinese cabbage with yellow or orange inner leaves are markedly higher than, and different from, those of common white leaf varieties.

CRTISO catalyzes the isomerization of poly-cis-carotenoids to all-trans-carotenoids. Together with PDS and ZDS, CRTISO is required to synthesize lycopene from phytoene. In *B. rapa*, the orange pigmentation of flowers and the inner leaves is under the control of the *BrCRTISO* gene. Using restriction fragment length polymorphism markers, the *orange–yellow* pigmentation gene (*Oy*) in Chinese cabbage was first mapped to linkage group 1 [26]. Linkage analysis finally located the candidate gene to the end of A09. Comparisons of the promoter regions of *CRTISO* promoter revealed insertions/deletions between the two parents, which identified *CRTISO* as the most likely candidate gene for *Br-Oy*. After using BC1 backcross population for high-quality mapping, three SNP markers delimited the *Br-Oy* locus was delimited to a 9.47 kb using three single nucleotide polymorphism (SNP) markers, which contained one functional gene, Bra031539 (*BrCRTISO*) [27]. The *Br-Oy* gene has many sequence variations, such as a 90-bp promoter deletion and a 501-bp 3′ end insertion. Meanwhile, a *CRTISO* mutation was found in an orange Chinese cabbage, which were used to develop molecular markers to differentiate orange from white genotypes [28]. Further study showed that these mutations in *BrCRTISO* reduced the flux of carotenoid synthesis, leading to the formation of orange leaves in Chinese cabbage. Genes involved in carotenoid metabolism were identified using white flowered varieties. According to genetic research, recessive loci *Brwf1* and *Brwf2* regulate Chinese cabbage’s white flowers. Insertion/deletion (InDel) and SNP tombstone analysis located *Brwf1* in a 49.6-kb region of chromosome A01 comprising nine annotated genes. *BrCRTISO* (Bra031539) was mapped to a 59.3-kb gap on chromosome A09 comprising 12 known genes and *Brwf2* [29]. To fine map the white flower gene *BrWF3* in Chinese cabbage, an F2 population was created from the F1 plants of a cross between a white flowered line and a yellow flowered line. *BrWF3* was precisely mapped to a 105.6 kb gap. Sequence variation analysis, functional elucidation, and expression profiling demonstrated that the *BrWF3* gene was most likely Bra032957, an *AtPES2* homolog. Carotenoid compound analysis and transmission electron microscopy revealed that *BrWF3* might produce xanthophyll esters, primarily violaxanthin esters, which interfere with chloroplast formation and plastoglobule (PG) generation. The third exon of *BrWF3* had an SNP deletion that prevented the protein from functioning properly and prevented PG assembly, accompanied by decreased expression of carotenoid metabolism-related genes [30]. In another study, a natural mutant of flowering Chinese cabbage (*B. rapa* ssp. *chinensis* var. *parachinensis*) with visually distinguishable pale-yellow petals was obtained in farmland. The pale-yellow petal was controlled by a single recessive gene *BrPYP* [31]. Further study showed that *BrPYP* was mapped to the same locus of *BrWF3*; however, different variations were identified. A functional 1148 bp deletion in the promoter region of *BrPYP* that reduces promoter activity and expression level was identified. In *B. napus*, mutation of *BnaA09.CRTISO* and *BnaC08.CRTISO* using the clustered regularly interspaced short palindromic repeats (CRISPR)/CRISPR associated protein 9 (CAS9) system caused the petals to turn milky white and the leaves to turn pale yellow. Thus, petal and leaf coloration in *B. napus* are regulated crucially and redundantly by *BnaA09.CRTISO* and *BnaC08.CRTISO*. The carotenoid concentrations in the petals and leaves of the *BnaCRTISO* double mutant were significantly decreased, according to subsequent observation. The levels of chalcone, a vital component of yellow color, were reduced dramatically in the mutant flower’s petals, yet its levels of lycopene, β-carotene, and α-carotene increased slightly [32]. These findings help explain how carotenoids are produced and how *B. napus*’s color variation is controlled. Chinese kale (*B. oleracea* var. *alboglabra*) contains abundant carotenoids, among which neoxanthin is one of the most important. When *BoaCRTISO* expression was downregulated in Chinese kale, virtually all carotenoid biosynthesis genes were downregulated, and the leaves turned yellow. The authors found that *BoaCRTISO* was a photoinduced gene, and a mixture of red, blue, and white light could increase the carotenoid content in plants [33]. Consequently, the CRISPR/Cas9 method was used to target and edit the Chinese kale *BoaCRTISO* gene. Decreases in the overall and individual carotenoid and chlorophyll levels were observed in the mutants. The color of the mutant changed from green to yellow, suggesting a reduced protective effect of carotenoids toward chlorophyll [34].

The hydroxylated β-rings of zeaxanthin were subjected to subsequent epoxidation by ZEAXANTHIN EPOXIDASE (ZEP) to produce antheraxanthin and later, violaxanthin. In addition to CRTISO, ZEPs were also identified to be widely involved in the coloration of flowers or leaves in different *Brassica* crops. In a study of the dark yellow petals of Chinese cabbage, bulked segregant RNA (BSR) sequencing combined with competitive allele-specific PCR (KASP) assays fine mapped *Br-dyp1* to a 53.6 kb region on chromosome A09. Further expression analysis, functional annotation, and sequence variation assessment identified Bra037130 (*BraA09.ZEP*) as a potential candidate gene for *Br-dyp1*. Plants with dark yellow petals had a 679 bp insertion in *BraA09.ZEP* that created a premature stop codon, leading to ZEP loss-of-function. Loss of ZEP activity destroyed the carotenoid metabolism, and increased accumulation of total carotenoid, and finally changed the petals from yellow to dark yellow of Chinese cabbage. Large amounts of violaxanthin lead to yellow petals in *B. napus*, while the orange flowers were caused by the accumulation of lutein. In *B. napus*, the ratio of yellow- to orange-flowered plants was 15:1 in the F2 population and 3:1 in the BC1 population, indicating that two dominant nuclear genes controlled the yellow-flower phenotype [35]. Map-based cloning was used to isolate key genes to reveal the underlying molecular mechanism of the orange-flowered phenotype. Quantitative trait locus (QTL) cloning assigned the change in color from yellow to orange to the loss of *BnaC09.ZEP* and the deletion of a 1695 bp fragment of *BnaA09.ZEP*. Further CRISPR/Cas9 and genetic complementation analyses showed that the nullification of *BnaA09.ZEP* and *BnaC09.ZEP* contributed to markedly increased lutein levels and a large decrease in violaxanthin levels in petals rather than in leaves [36].

CCDs encompass a superfamily of mononuclear non-heme iron proteins that catalyze the oxygenolytic splitting of alkene bonds in carotenoids to develop apocarotenoid products. β-carotenoid cleavage dioxygenase 4 (CCD4) was reported to be the primary negative switch for seed carotenoids, particularly β-carotene, according to linkage mapping and genome-wide association studies of *Arabidopsis* carotenoids [37]. In *B. napus*, positional cloning identified a carotenoid cleavage dioxygenase 4 gene, *BnaC3.CCD4*, as being responsible for the composition of white or yellow flower color, with white-bloomed *B. napus* lines having higher expression of *BnaC3.CCD4* in their petals. In yellow-flowered varieties, a CACTA-like transposable element 1 (TE1) is embedded in the coding region of *BnaC3.CCD4*, which disrupts *BnaC3.CCD4* expression. Further investigation uncovered that this TE insertion occurred frequently in the *BnaC3.CCD4* gene of yellow-flowered *B. napus* [38]. In *B. oleracea*, analyses demonstrated that this yellow-white petal characteristic was dependent a single locus on C03, and in 2019, Han mapped the gene to a 207-kb locus, suggesting that the candidate gene was *BoCCD4*. Further sequence analysis, expression pattern assessment, and functional complementation analyses in *B. oleracea* accessions showed that *BoCCD4* functional failure resulted the yellow petal phenotype. Overexpression of *BoCCD4* changed the color of the petals from yellow to white or pale yellow [39]. In cauliflower, the yellow-flower locus was fine-mapped, which identified *BoCCD4* as the most likely candidate gene. Further investigation revealed the presence of a novel 10,608 bp CACTA-like transposon that inhibited the function of *BoCCD4* [40]. The yellow-petal feature could be induced in a white-petal natural line by *BoCCD4* functional complementation. *BoCCD4* was observed to be particularly expressed in the petal tissue of white-petal plants, and a genetic study revealed that in carotenoid metabolism, CCD4 homologs might share evolutionarily conserved functions. BoAAO3 is a key enzyme that interacts with *BoCCD4* to regulate petal carotenoid deterioration. Likewise, *BoCCD4* co-regulates carotenoid metabolism accompanied by two key transcription components, Bo2g151880 (WRKY) and Bo3g024180 (SBP). Together, they regulate carotenoid biosynthesis in petals, which in turn adjusts whether petals are white or yellow [41].

The bioavailability of carotenoids from new and processed snacks heavily depends on their natural deposition form. In cauliflower (*B. oleracea* var *botrytis*), a spontaneous semidominant *Orange* (*OR*) mutant has a fascinating genetic mutation that results in the accumulation of β-carotene in commonly unpigmented tissues. Using positional cloning, the gene responsible for *OR* was determined, and functional complementation confirmed this identification in wild-type cauliflower. *OR* encodes a DnaJ Cys-rich domain-containing plastid-associated protein. The *OR* allele contains an inserted long terminal repeat retrotransposon, resulting in the *OR* gene mutation. Analyses of the gene, its output, and the effects of an *OR* transgene on cells indicated that *OR*’s function is associated with a biological process that promotes the differentiation of proplastids or other noncolored plastids into chromoplasts for carotene accumulation. Additionally, the study demonstrated that that regulating chromoplast formation is essential for the control of carotenoid production in plants [42]. Regarding leafy *Brassica* crops, cultivars with golden leafy heads are becoming increasingly recognized. The golden cultivars are abundant in β-carotene and lutein. In comparison with the white line, the β-carotene level was increased by 13.6−fold. Bulked-segregant study sequencing identified *BraA09g007080.3C* (encoding the ORANGE protein) as the candidate gene. A 4.67 kb long terminal repeat was observed to be inserted in exon three of *BrGOLDEN*, which resulted in the expression of three alternatively spliced transcripts. Spatiotemporal expression analysis showed that *BrGOLDEN* might affect the expression levels of carotenoid-related genes [43]. Further sequence analysis revealed that *BrGOLDEN* was probably transferred into *B. rapa* through distant hybridization between *B. rapa* and *B. oleracea*. In addition, in the first step of carotenoid biosynthesis, orange protein (OR) is known to interact with phytoene synthase (PSY) and is a major post-transcriptional regulator on PSY in *Arabidopsis* [44].

Additionally, turnip (*B. rapa* ssp. rapa) is a nutritious and fitness-promoting vegetable, with yellow and white flesh, among which yellow-fleshed turnips have a higher nutritional value. Combined transcriptomic and metabolomic investigations identified that *PSY* is the important gene that affects carotenoid creation in turnip. High expression of *PSY* results in yellow turnips rather than mutations in *PSY*. This suggested that carotenoids might be produced via a post-transcriptional regulatory mechanism.

In addition to *CRTISO*, *ZEP*, *CCD*, and *OR*, certain other genes were reported to be involved in carotenoid absorption. A mutant with *yellowish-white* flowers (*ywf*), which was developed from Zhongshuang 9 (ZS9) using ethyl methane sulfonate mutagenesis was analyzed. The *ywf* locus displayed a lower petal carotenoid content. A genetic study revealed that a single recessive gene regulated the yellowish-white trait. Bulked segregant analysis sequencing mapped the *ywf* locus to *YWF*, encoding phytoene desaturase 3 (PDS3). Moreover, *ywf* accommodated a C-to-T replacement in the coding region, which caused premature translation termination. RNA-seq and carotenoid constituent investigation showed that in *ywf* petals, the truncated *BnaA08.PDS3* disrupted carotenoid biosynthesis [45]. In kale, the neoxanthin synthase gene (*BoaNXS*) functions to adjust leaf color, and in *BoaNXS* overexpressing plants, the color alternated from yellow-green to green, and the total carotenoids and individual carotenoid contents were considerably elevated [46].

## 4. Evolution of Carotenoid Biosynthesis and Some Key Carotenoid Genes in *Brassica*

The consumption of *Brassica* species confers unique health attributes, and they contain high carotenoid levels. The six genetically linked *Brassica* species share a common evolutionary history and are currently being bred using interspecific hybridization for species improvement [47]. In this background, to increase the nutritional value of *Brassica*, identifying the genetic connections among *Brassica* species related to carotenoid accumulation would promote their genetic improvement. The main carotenoids that accumulate in genetically-related *Brassica* species have been identified. *B. rapa* was reported to accumulate the highest concentrations of antheraxanthin, lutein, and zeaxanthin. The maximum concentrations of β-carotene and total chlorophyll were detected in *B. juncea. B. nigra* contained the highest levels of 5,6-epoxylutein and violaxanthin, while *B. oleracea* had the highest neoxanthin levels. Interestingly, the amphidiploids *B. carinata* and *B. napus* were found to contain significantly reduced levels of carotenoids compared with those in the diploid species and *B. juncea* [8].

Plant whole-genome triplication (WGT) resulted in multiple duplicates of carotenoid biosynthetic genes, most of which retain their syntenic relationships with their *A. thaliana* orthologs. Taking *B. rapa* as an example, *B. rapa* diverged from *A. thaliana* and its nucleus contains three subgenomes [48]. Gene loss events in the three subgenomes of *B. rapa* show bias. However, the proportions of carotenoid biosynthetic genes in the respective *B. rapa* subgenomes were not considerably different from those in the whole genome. Flower color is mostly determined by the presence or deficiency of carotenoid pigments. The production of carotenoids in chromoplasts of petal cells leads to the yellow color of petals. The prevailing flower color of *Brassica* spp. according to the triangle of U hypothesis is yellow. However, among the subspecies of the *B. oleracea* cytodeme, some white-flowered varieties are observed. Moreover, white flowers were reported in rapeseed lines created via interspecific hybridization between white-flowered *B. oleracea* and *B. rapa*. Herein, we used *B. napus* as an example to explain the evolution of carotenoids in *Brassica*. *B. napus* (2n = 4 = 38, AACC) is an allopolyploid crop plant formed by an interspecific crosses between *B. rapa* (2n = 2 = 20, AA) and *B. oleracea* (2n = 2 = 18, CC). Flower pigment divergence was evident in the selfed offspring from a single *B. napus* parent, varying from white to bright yellow. This variation was assumed to be genetically related because all offspring were identified under controlled environmental conditions. In *B. napus* and *B. oleracea*, the *carotenoid cleavage dioxygenase 4* gene (*CCD4*) exhibits the white flower trait. Among *B. carinata*, *B. rapa*, and *B. oleracea*, five distinct alleles of *C3.CCD4* were identified. The wild-type (WT) allele was designated as that contained by white-flowered lines of *B. rapa* and *B. oleracea*. The four variant alleles comprised two InDels (M2 and M3) and two with TE insertions (M1 and M4), and were contained in yellow-flowered *Brassica*s: *B. oleracea* and the allotetraploids *B. napus* and *B. carinata*. The color of the petals changed from white to yellow as a result of these variant alleles disrupting *C3.CCD4*’s function. Interestingly, all yellow-flowered *B. napus* and *B. carinata* plants and some yellow-flowered *B. oleracea* plants are homozygous for M1 or M4, suggesting that the CACTA-like TE insertion into *C3.CCD4* occurred before *B. napus* and *B. carinata* allopolyploidization. The two InDels were only detected in *B. oleracea BolC3.CCD4*, but not in the *CCD4* genes of *B. napus* or *B. carinata*. This suggested that mutations M2 and M3 occurred outside of the centers of origin of *B. napus* and *B. carinata*, making no contribution to their speciation. Using *AtCCD4* as the outgroup, a phylogenetic tree of the five *Brassica C3.CCD4* alleles was constructed, which suggested that the WT allele was the ancestral and functional type, and the others were loss-of-function types. This prompted the conclusion that the *B. oleracea* flower color diversified into yellow and white flowers prior to the emergence of the amphidiploids *B. napus* and *B. carinata*, resulting from CACTA-like TE insertions in the *B. oleracea BolC3.CCD4* gene. Thus, we presumed that the development of blossom color in *B. carinata*, *B. napus*, and *B. oleracea* is governed by the evolution of *CCD4* in the *Brassica* C genome [38].

In addition, the dynamic expression pattern of genes played a substantial role in the development of carotenoid biosynthesis during evolution. *Brassica* diploids were found to contain two ZEP homologs, while *Brassica* allotetraploids were found to have four. Genetic analyses allowed the assignment of *Brassica* ZEPs to subclades based on their localization to the A or C genome. ZEPs translated from genes located on chromosomeA07/C07 formed a subclade that was most closely related to *Arabidopsis* ZEP proteins, forming a sister cluster to the ZEPs located on *Brassica* chromosomes A09/C09. These results suggested that *Brassica* ZEPs diverged prior to the allopolyploidization event. Furthermore, *BnaA09.ZEP* and *BnaC09.ZEP* were reported to be mostly expressed in floral tissues, while homologous *BnaA07.ZEP* and *BnaC07.ZEP* were mainly expressed in leaves. These observations revealed that *BnaZEPs* were redundant and experienced tissue-specific diversification [36]. These discoveries, taken together with earlier data for CCD4, revealed the central role exerted by gene duplication and tissue-specific expression in the evolution of carotenoid accumulation in *Brassica*. Indeed, accumulating evidence demonstrated that the evolution of tissue-specific expression following gene duplication might be unique to plant carotenoid metabolism. In tomatoes, many carotenoid-metabolic genes, such as GGPPS, PSY, LCYB, and BCH, are present in multiple copies: one copy is preferentially expressed in green tissues, and the other in flowers or fruit.

## 5. Biotechnological Implications of Carotenoid Genes Identified by QTL-Mapping

Research has sought to develop crops with higher carotenoid contents because of their status as essential phytonutrients. Vegetable yield and quality are affected by cultivars, harvesting dates, climate, location, and conditions. However, conventional methods to improve quality, e.g., chemical spraying and hybridization, are often lengthy and ineffective. Plant disease resistance and yield have been improved using the efficient and convenient CRISPR/Cas9 system [49]. Such technology will drive innovative strategies for metabolic engineering to produce crops with higher nutritional value. Improved crop varieties with modified carotenoid pathways will have a wide application in horticulture.

The breeding of plants with different colors has been facilitated by the cloning and functional analysis of flower pigment regulatory gene. Indeed, certain transgenic ornamental plant varieties with modified flower color have been developed. Agro-transformation of *B. napus* has successfully altered its flower color. For example, *B. napus* with yellow flowers was transformed with *PAP2* (encoding phytochrome-associated protein 2) controlled by a petal-specific promoter, which increased the petal anthocyanin content, generating red flowers [50]. Moreover, both *BnaA09.ZEP* and *BnaC09.ZEP* were mutated using CRISPR/Cas9, which changed the profile of carotenoids in the petals of the mutant plants, without disturbing their growth and development. The CRISPR/Cas9 system was also used to edit *B. napus BnaA09.CRTISO* and *BnaC08.CRTISO*, resulting in a mutant phenotype of yellowish leaves and creamy white petals in the double mutant plants. The carotenoid isomerase gene (*BoaCRTISO*) of Chinese kale was also targeted and edited, which decreased the chlorophyll content and the total and individual carotenoid levels. These studies showed that the manipulation of known genes that are genetically determined by QTL mapping represents an auspicious approach to alter the appearance of flowers and to breed *Brassica* crop varieties with different colored flowers with high ornamental worth.

## 6. Conclusions and Perspective

Consumption of carotenoids has been associated with various health benefits, including a reduced risk of age-related macular degeneration and cataract, some cancers and coronary heart disease [51]. There is also some evidence of a beneficial effect on cognitive function [52]. The content and composition of carotenoids have vital functions in the nutrition and quality of *Brassica* crops [53]. However, humans cannot synthesize carotenoids and must ingest them in food or via supplementation. In recent times, large numbers of genes and the mechanisms that regulate carotenoid biosynthesis and accumulation have been discovered (Figure 1 and Table 1), facilitating the smooth breeding of *Brassica* crop [54]. Here, we reviewed recent progress regarding carotenoids of *Brassica* from the perspective of forward genetics. We believe that this review provides biotechnological implications and some new perspectives on carotenoid research in *Brassica*, as well as other horticultural crops. However, the regulation of carotenoids in higher plants is complex and multi-faceted; thus, more effort should be made to elucidate the mechanisms of carotenoid function.

## Figures and Tables

**Figure 1 plants-12-01117-f001:**
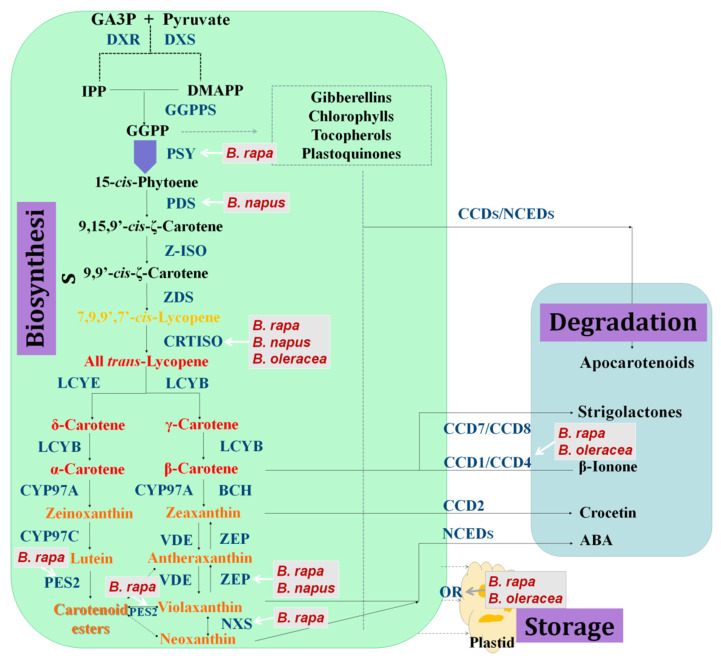
Overview of carotenoid metabolism in *Brassica*.

**Table 1 plants-12-01117-t001:** Enzymes and genes that regulate carotenoid accumulation in *Brassica* crops.

Regulated Genes	Species	Major Changes	Genes (Gene Accession)	Color Change/Tissue	Reference
*CRTISO*	*B. rapa*	The mutant Br-oy protein cannot convert prolycopene to all- trans -lycopene.	*BrOy* (*Bra031539*)	white/yellow → orange inner leaf	[26,27]
		The loss of BrCRTISO function leads to the accumulation of prolycopene.	*Br-oy or BrCRTISO* (*Bra031539*)		
		Loss of BrWF3 function interferes with plastoglobules assembly and decreases expression levels of genes associated with carotenoid metabolism.	*Brwf3 * (*Bra032957*)	yellow → white/flower petal	[30]
		The key factor for the pale-yellow color of petals was the decrease in esterified carotenoid content due to the loss of PYP function.	*BrPYP* (*BraA02g037170.3C*)	yellow → pale-yellow/flower petal	[31]
	*B. napus*	The contents of carotenoids in petals and leaves of BnaCRTISO double mutant were reduced. In petals, the content of chalcone decreased, the content of some carotene (lycopene, α-carotene, γ-carotene) increased.	*BnaA09. CRTISO* (*BnaA09g49740D*) *BnaC08. CRTISO* (*BnaC08g44970D*)	yellow → milky white flower petals yellow → pale yellow leaves	[32]
	*B.oleracea*	Carotenoid and chlorophyll levels were reduced in the mutant of BoaCRTISO.	*BoaCRTISO * (*GenBank accession MN810158*)	green → yellowing leaves	[33,34]
*ZEP*	*B. rapa*	The loss of function of ZEP disrupts the metabolism of carotenoids and leads to the increase in total carotenoid accumulation.	*Br-dyp1* (*Bra037130*)	yellow → dark yellow flower petal	[35]
	*B. napus*	The abolishment of both genes led to a substantial increase in lutein content and a sharp decline in violaxanthin content in petals.	*BnaA09. ZEP * (*BnaA09g07610D*) *BnaC09. ZEP * (*BnaC07g16350D*)	yellow → orange flower petal	[36]
*CCD4*	*B. napus*	In yellow petals, a large amount of α-carotene, α-cryptoxanthin, β-cryptoxanthin, violaxanthin, 9-*cis*-violaxanthin, lutein, and *cis*-neoxanthinwere accumulated.	*BnaC3.CCD4* (*Bol029878*)	white → yellow flower petal	[38]
	*B.oleracea*	Not available.	*BoCCD4* (*Bol029878*)	white/pale yellow → yellow flower petal	[39]
		These key genes may interact with *BoCCD4* to jointly regulate carotenoid biosynthesis in petals.	*WRKY* (*Bo2g151880*) *SBP* (*Bo3g024180*)		[41]
*OR*	*B.oleracea*	The *OR* gene mutation confers the accumulation of high levels of β-carotene in various tissues normally devoid of carotenoids.	*OR * (*GenBank accession DQ482460*)	white → orange	[42]
	*B. rapa*	The *BrGOLDEN* lines are rich in β-carotene and lutein.	*BrGOLDEN* (*BraA09g007080.3C*)	golden → light yellow inner leaf	[43]

## Data Availability

The data used to support the study are available from the corresponding author upon request.

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
