# Peer review of "Recent Advancements and Biotechnological Implications of Carotenoid Metabolism of Brassica"

_plants, 2023, doi:10.3390/plants12051117_

Round 1

Reviewer 1 Report

- In fact, the title of the research attracted me to know its content, but when I started reading, I became bored and frustrated. The topic was presented and dealt with in this way by the researchers. It was not good, as there are no illustrations to explain all the scientific talk and scientific abbreviations that exist. I think in this way the research does not amount to publication in a classified journal in Q1 and specialized journal that has a high impact factor.

- There are many missing references, especially since it is a review

- The researcher choose only one crop as an example, which is Chinese red cabbage, while the family that was tested has many other crops, so it was necessary to choose several examples and make comparisons between them to confirm the point of view and the strategies that were discussed.

- Some words were used that could not be used in scientific research such as People 

-I do not advise the acceptance of this review for publication in this way, so the researchers must revise it and make it better in order to attract readers to benefit from it and understand it

Reviewer 2 Report

Brassica is one of the most important vegetable genera worldwide. In this review, the authors did a quite comprehensive update of the current understanding of carotenoid metabolism and regulation in Brassica species. The overall quality of the review is high. This reviewer has only a few minor suggestions.

1. Line 34. “circulated" is not correct. It is distributed but not circulated.

2. Line 39. "double bonds that can absorb triplet state chlorophyll" is wrong. Double bonds can absorb energy, but not chlorophyll.

3. Line 41. "pollinating insects". Don't forget there are also animals for seed dispersal attracted by the color and fragrance (ionones, etc.) of fruits.

4. Line 48. Golden Rice should be cited here.

5. Line 65. MVA pathway does not directly contribute to carotenoid metabolism.

6. Figure 1. Please remove the round dot at another end of the arrows. They look like arrows as well in the figure. Moreover, OR is known to interact with PSY. Please place it accordingly.

7. Line 199. "Loss of ZEP activity destroyed the lutein cycle and the lutein epoxy cycle" is confusing. ZEP is not involved in lutein metabolism in Brassica.

8. Line 241. "botrytis" should in italic.

9. Line 253. "leafy heads" should be "curds".

10. Line 254. The golden cultivar of cauliflower accumulates beta-carotene but not lutein.

11. Lines 423, 429, 441, 478, 483. The authors' names are incorrect. Line 419 also misses page numbers. Please also check other references.

Round 2

Reviewer 1 Report

-It is clear that the authors did not understand my first point of view, and that my goal is to improve their manuscript so that the reader does not get bored with it and the topic becomes interesting to read. Therefore, I will encourage them with these suggestions, and I hope that they will improve their manuscript more.

- Please add more than one illustrative figures to more details and to reduce To reduce boredom from scientific explanation a lot 

- I saw that the authors added some references, but I ask them to add more, which improves and highlights the effort made
